# Intervention to Increase Cervical Cancer Screening Behavior among Medically Underserved Women: Effectiveness of 3R Communication Model

**DOI:** 10.3390/healthcare11091323

**Published:** 2023-05-05

**Authors:** Matthew Asare, Anjelica Elizondo, Mina Dwumfour-Poku, Carlos Mena, Mariela Gutierrez, Hadii M. Mamudu

**Affiliations:** 1Department of Public Health, Robbins College of Health and Human Sciences, Baylor University, Waco, TX 76798, USA; anjelica_elizondo@baylor.edu (A.E.); mina_dwumfour-poku1@alumni.baylor.edu (M.D.-P.); 2North American University, Stafford, TX 77477, USA; cjmv21@gmail.com; 3Department of Population Health and Health Disparities, School of Public and Population Health, University of Texas Medical Branch at Galveston, Galveston, TX 77555, USA; maraguti@utmb.edu; 4College of Public Health, East Tennessee State University, Johnson City, TN 37614, USA; mamudu@etsu.edu; 5Center for Cardiovascular Risk Research, East Tennessee State University, Johnson City, TN 37614, USA

**Keywords:** 3R communication model, self-sampling, medically underserved women, cervical cancer screening

## Abstract

Human Papillomavirus (HPV) self-sampling has the potential to increase Cervical Cancer Screening (CCS) and reduce the cervical cancer burden in Medically Underserved Women (MUW). However, interventions promoting self-sampling are limited. We examined the effectiveness of an intervention study in increasing CCS among MUW. We conducted a quasi-experimental intervention study. A face-to-face verbal approach was used to recruit MUW (*n* = 83, mean age 48.57 ± 11.02) living in a small city in the US. Behavioral intervention based on reframing, reprioritizing, and reforming (3R model) was used to educate the women about CCS in a group format. The women (*n* = 83) completed pre-and post-intervention assessments, and 10 of them were invited for follow-up interviews. The primary outcome was CCS uptake. Mixed methods analyses were conducted using a *t*-test for the primary outcome, PROCESS for mediation analysis, and NVivo for interview data. The majority of women (75%) completed self-testing. High-risk HPV among women was 11%, and of those, 57% followed up with physicians for care. We found that the significant increase in the women’s post-intervention screening behaviors was mediated by the increase in knowledge (Indirect Effect [IE] = 0.1314; 95% CI, 0.0104, 0.4079) and attitude (IE = 0.2167; 95% CI, 0.0291, 0.6050) scores, (*p* < 0.001). Interview analyses offered further explanations why MUW found the intervention messages acceptable (encourages proactive behavior), feasible (simple and easy to understand), and appropriate (helpful and informative). Barriers, including lack of trust and fear of results, were identified. The findings suggest that an intervention that combines the 3R model and self-sampling may increase CCS among MUW.

## 1. Introduction

The significant decrease in cervical cancer death rate over the past three decades in high-resource countries is one of the public health success stories [1]. This decline is due to several factors, including the introduction of Cervical Cancer Screening (CCS) and Human Papillomavirus (HPV) vaccination, improvements in treatment, and successful cytology-based screening programs [1]. However, the disease remains a significant public health threat, especially for Low-Income Women (LIW) in the United States (US). As of 2020, there were about 296,981 women who were living with cervical cancer in the US [2]. The age-adjusted rates for new cases and mortality of cervical cancer were 7.7 and 2.2 per 100,000 women per year [2], respectively. Approximately 14,000 women are diagnosed with invasive cervical cancer in the US, and 4290 patients die from the disease each year [3,4]. Additionally, disparities exist in cervical cancer incidence among different racial and ethnic groups in the US, with a much higher incidence of new cases occurring among Black and Hispanic women compared to non-Hispanic white women [5,6,7]. The cervical cancer incidence rates among US Hispanic and non-Hispanic black women are nearly 40% [8] and 30%, [9] respectively, higher than among non-Hispanic whites [5,6,7]. Evidence shows disparities in the detection and survival of cervical cancer between African Americans and Whites [10,11], between women with low socioeconomic status and women with higher socioeconomic status [10], and between uninsured and Medicaid-insured persons compared with privately insured persons [10,12,13]. Disparities in cervical cancer health outcomes are preventable because cervical cancer is easily detected, the means for detection are inexpensive, and treatment is effective if the disease is detected in its early stages [10,14]. The most common risk factor for persistent cervical cancer cases is non-participation in cervical cancer screening [15,16,17]. Therefore, reducing underscreening among women is a key prevention priority, as more than 50% of the cervical cancers diagnosed annually are in under-screened women [18,19].

### 1.1. Disparities in Cervical Cancer Screening

Fortunately, screening devices such as HPV tests, pap tests, and Visual Inspection with Acetic Acid (VIA) have been developed for the early detection of cervical cancer. Cervical cancer is curable if detected at an early stage. National organizations such as the U.S. Preventive Services Task Force (USPSTF), the American Cancer Society (ACS), and the American College of Obstetrics & Gynecology (ACOG) have issued various recommendations for cervical cancer screening. While there are variations in the cervical cancer screening guidelines among the various agencies, there is a near consensus among the agencies that women aged 21–29 are recommended to have a pap test alone every 3 years, and women aged 30–65 should have a pap test alone every 3 years, or HPV test alone every 5 years, or pap test and HPV co-testing every 5 years [20]. Currently, many pap tests (cytology) require a physician to obtain samples from the cervix for further examination, while HPV tests require samples from the cervix but can be obtained using brushes or swabs, or other devices by either physician or by screening participants. However, the utilization of cervical cancer screening remains suboptimal for approximately 3 out of 10 women [21]. In 2019, the overall cervical cancer screening rate of 73.5% among US women was below the Healthy People 2030 goal of 84.3% [21]. Additionally, race, income, and geographical location disparities exist in cervical cancer screening. Women that are less likely to be screened have lower socioeconomic status and educational attainment, are racial/ethnic minorities and foreign-born, are residents of rural counties, areas with persistent poverty, and areas with geographic inaccessibility to adequate screening services [4,21,22]. For instance, low-income women are under-screened compared to high-income women. Screening rates in high-income women are 87% compared to just 66% in low-income women [23]. Non-Hispanic white women are more likely to get cervical cancer screening than Black or Hispanic women [23]. Depending on geographical areas, the screening rates can be lower than the national average. For instance, the southern part of the US, including Texas, has lower screening rates than the national average rates [24,25]. Our study among 254 LIW receiving food from a food pantry in central Texas showed that only 54.8% were current on their screening [26]. Several studies have come to a similar conclusion that the screening rate among LIW in the US is 66% [23,27].

### 1.2. Barriers to Cervical Cancer Screening

There are several barriers to cervical cancer screening. The barriers include cost, fear of finding cancer, anxiety, embarrassment, the anticipation of pain, male physician presence, lack of knowledge about screening and misinformation among those who are aware of screening, language barriers, other health issues, transportation, forgetting to schedule appointments, and lack of time to go for screening [25,28,29,30,31,32,33].

A major way of addressing some of the barriers is using self-sampling, a method where women collect vaginal samples themselves and send them to the clinic or laboratory for analysis. Self-screening approaches may be acceptable, relatively easy to implement, and cost-effective to be sustainable. Offering women the option to self-collect vaginal or cervical samples at home could likely increase participation in cervical cancer screening programs [34]. Self-sampling could reduce the potential financial [35] and logistical burden for the patient and allows for a greater initial sense of privacy and autonomy. Self-sampling can increase access to cancer screening for women who live long distances from medical facilities that provide in-office screening, have difficulty attending appointments due to transportation challenges or work/caregiving responsibilities, are uncomfortable in medical settings or with healthcare providers, prefer to avoid pelvic exams (e.g., due to cultural/religious preferences or history of sexual trauma) [4,22]. Evidence shows that self-sampling is efficacious in detecting precancerous lesions and could address most screening-related barriers if adopted in low-resource areas [36,37]. Studies showed that women were more likely to report a positive experience with self-sampling and showed higher participation rates in self-sampling than in physician-performed pap smears and HPV co-testing [36,38]. Meta-analyses of data from observational studies and randomized controlled trials show HPV self-sampling-based tests have (a) higher sensitivity compared to cytology, (b) comparable sensitivity vs. clinician-collected sampling for polymerase chain reaction (PCR)-based HPV DNA assays, and (c) high positive agreement vs. clinician-collected sampling for PCR-based HPV DNA assays [39,40,41,42]. Notwithstanding, the implementation of self-screening remains poor in many low-resource areas, partly because most women are unaware of the self-screening option. Additionally, there is no structured and consistent message and communication model to deliver self-screening messages. The 2022 President’s cancer panel report recommended that the public and healthcare providers must have accurate, digestible, and actionable information about cancer screening. Therefore, it is critical to develop robust communication models and social mobilization programs, as well as evidence-based implementation strategies, to enhance screening acceptance and utilization [43]. We developed an intervention based on the 3R (Reframing, Reprioritizing, and Reforming) communication model [44] to promote self-sampling among low-income women. We also determined whether the Theory of Planned Behavior (TPB) constructs (attitude and perceived behavioral control, self-efficacy) and knowledge would explain the women’s screening behavior.

### 1.3. Theoretical Framework

3R Communication model. The 3R model is based on a communication framework that seeks to reframe health information, reprioritize the information, and reform behavior about health information [44,45]. The 3R communication model provides a framework to simplify and structure cervical cancer screening messages. Reframing concepts are based on gain and loss-framed health information strategies [46,47]. The screening information emphasizes the costs of failing to screen (i.e., a loss-framed appeal) vs. the benefits of screening (i.e., a gain-framed appeal) [46,47]. Reprioritizing argues for making a given health behavior (i.e., screening) a priority to forestall the future burden of the health problem. Reforming focuses on helping individuals to develop a positive attitude toward the behavior (i.e., screening) as well as demystifying misconceptions about the health behavior (i.e., self-sampling). The 3R model messages have been shown to be effective in overcoming stigma and religious objections associated with mammogram usage [44] and intention for HPV vaccination acceptance [45].

The Theory of Planned Behavior (TPB) was developed to predict human behavior [48], and the TPB construct of behavioral intention explains motivational factors that influence behavior. The stronger a person’s intention towards engaging in a given behavior (i.e., self-screening), the more likely a person is to perform that behavior. The second construct is the attitude towards behavior which explains that a person may have a favorable or unfavorable appraisal of a given behavior (i.e., self-screening). Two components of attitude toward a behavior are behavioral beliefs and outcome evaluations. The third construct is the subjective norm, defined as a social pressure to perform or not perform a given behavior. Two components of the subjective norm are normative beliefs and motivation to comply. The final construct is perceived behavioral control, explained as a person’s perception of the ease or difficulty of performing the behavior of interest (i.e., self-screening). To our knowledge, no study has used these psychosocial factors (TPB constructs) to understand self-sampling. The primary purpose of the intervention was to assess the preliminary efficacy of the 3R communication model on increasing cervical screening uptake, defined as the completion and return of the sampling kit. Secondary outcomes were the acceptability, feasibility, and appropriateness of the 3R communication model and self-sampling. We also examined whether the 3R model changed participants’ knowledge, attitude, and perceived behavioral control (confidence) about cervical cancer and if that change mediated (influenced) their cervical cancer screening behavior.

## 2. Materials and Methods

### 2.1. Study Design and Recruitment

We conducted a quasi-experimental single-group pre- and post-community-based intervention design study. Eligible participants were generally healthy women (female-identified at birth) who were at increased risk for cervical cancer as defined by one of the following: between 30 and 65 years old and had never had a pap smear or HPV test before or had not had cytology alone for the past three years or had not had cytology and HPV co-testing for the past five years. Women who can read and write in English and/or Spanish and can give consent per Institutional Review Board stipulations were included in the study. Women who were less than 30 years or older than 65 years, had had cytology in the past three years or had had cytology plus HPV testing for the past five years, were pregnant, or had a hysterectomy were not eligible. We used face-to-face and snowball methods to recruit a purposive sample of low-income women from the community in a small southern city in the US.

(a)The face-to-face method was used to recruit some of the women at community gatherings such as local food pantries and churches. During our first contact with the potential participants, we gave them the study recruitment flyer which had the study eligibility criteria (in English and Spanish) and our contact information. Upon reading it, some of them instantly informed us of their willingness to participate in the study and gave us their phone number. Others took the flyers with them and made decisions afterward. Women were recruited once initial inclusion qualifications were determined.(b)The snowball method was used when a woman completed the study; we asked her if she would like to introduce anybody, including friend(s), family member(s), or co-worker(s), to the study. Some of the participants offered to introduce the study to women in their network. When we received the contact information of the women referred, we followed up with them and assessed their eligibility based on the study’s inclusion/exclusion criteria. In both recruitment methods, we contacted the women through the phone numbers they gave to us, and once their eligibility had been determined, we discussed informed consent with them and scheduled an intervention presentation time for those who qualified and were willing to participate. We had a designated facility in the community area where the presentations were conducted. We gave the address of the facility to the women, and they drove to the facility on their scheduled date. We provided transportation to those women who did not have access to transportation. The study protocol was approved by the Institutional Review Board of the University.

### 2.2. Intervention Description and Delivery

The intervention consisted of three modules and each module covered an area of the 3R framework. Module 1 covered the reframing concept. Basic information about HPV, HPV-related cancers, and the disease health implications was presented to the participants. The reframe emphasized the costs of failing to screen (i.e., a loss-framed appeal) vs. the benefits of screening (i.e., a gain-framed appeal) [44,45]. Additionally, we reframed that screening is a mechanism for early detection of cancer risk as opposed to late diagnosis of cancer. We explained that screening for early detection can lead to finding a solution to prevent the precancerous condition of the cervix (i.e., carcinoma in situ) from becoming cancerous.

Module 2 focused on the reprioritizing concept. Basic information about cervical cancer prevention, including screening, screening types, the importance of screening, and the implications of failure to screen, was presented to the participants. We emphasized the cost of screening now vs. the cost of cancer treatment. We educated them that cervical cancer is preventable and that prioritizing screening participation is an important step toward prevention.

Module 3 addressed the reforming concept. Basic information about myths and misinformation about screening and cervical cancer was presented to the participants. The presentation emphasized the development of a positive attitude among LIW about screening as well as demystifying misconceptions about self-sampling. The reforming concept addressed the fear of detecting cancer. We explained that generally, the main purpose of following proper screening protocol is to determine whether women are predisposed to having cancer or are at risk of cancer.

### 2.3. Intervention Delivery

We organized a series of intervention presentations for the women, and each presentation was between 30 and 45 min long, depending on the number of questions the women asked. The presentations were in a group face-to-face PowerPoint format, and in each group, there were a minimum of two participants and a maximum of 10 participants. The PowerPoint presentation gave the women basic information about cervical cancer, and we followed it up with discussions centered on the 3R model approach to invite the women to reflect deeply on what the information they just received means to them now and in their future. After that, we opened the presentation up for questions and answers. Before the intervention, each woman, who agreed to participate, gave verbal informed consent and completed the hard copy of the baseline assessments before starting the intervention presentation.

### 2.4. Measures

Medical records: The primary outcome was self-sampling, defined as the completion and return of a self-sampling kit. The primary outcome was assessed using the medical records from the lab after the intervention.

Survey: The secondary outcomes were changes in pre-intervention and post-intervention scores for knowledge about self-sampling and cervical cancer, attitude towards self-sampling, and perceived behavioral control for self-sampling. Participants completed baseline and post-intervention assessments based on a validated Theory of Planned Behavior instrument [49].

Attitude subscale: The two constructs for the attitude towards behavior subscale were behavioral beliefs (2 items) and outcome evaluation (2 items). The behavioral belief items were (a) “Taking the HPV self-sampling test will result in knowing my health status about my chances of getting or not getting cervical cancer” and (b) “Taking the HPV self-sampling test will help me obtain information regarding my cervical cancer status and help plan for my future healthcare”. A 7-point Likert scale starting from “less likely” to “most likely” was used. The items for outcome evaluation were (a) “Taking the HPV self-sampling test to know my status with regards to cervical cancer is …… to me,” and (b) Taking the HPV self-sampling test to obtain information regarding my cervical cancer status and help plan for my future health care is … to me. Both items were measured on a 7-Likert scale starting from “extremely unimportant” to “extremely important.” The internal consistency for the attitude towards behavior subscale was calculated and the Cronbach’s alpha for the attitude towards behavior subscale was 0.89 [49].

Perceived behavioral control subscale: The two constructs for the perceived behavioral control (confidence) subscale were control belief (2 items) and influence on control belief (2 items). The control belief items were (a) I am confident that I can take the HPV self-sampling test and (b) It will be difficult for me to take the HPV self-sampling test. The items were measured using a 7-point Likert scale starting from “strongly disagree” to “strongly agree.” The perceived power items were (a) If I feel confident in my ability to take the HPV self-sampling test, I will be … take it and (b) If I feel that the HPV self-sampling test is difficult for me to take, I will be … take it, all measuring on a 7-point Likert scale starting from “less likely” to “most likely.” The internal consistency for the perceived behavioral control subscale was calculated and the Cronbach’s alpha for the perceived behavioral control subscale was 0.73 [49].

Knowledge subscale: Three items were used to assess participants’ knowledge and they were (a) “Human papillomavirus (HPV) can cause cervical cancer”, (b) “Cervical cancer can be prevented”, and (c) “Cervical cancer screening can help reduce the risk of cervical cancer”. The 7-point Likert scale response for the knowledge items ranged from “strongly disagree” to “strongly agree”.

Other covariates: Demographic variables included participant age, race/ethnicity, marital status, education, insurance, employment, annual income, screening behavior, feasibility (3 items), acceptability (3 items), and appropriateness (3 items).

Pre-assessment: Each woman individually completed a self-report questionnaire at the baseline. A hard copy of the questionnaire assessing the participants’ demographic information, knowledge, attitude, and perceived behavioral control for self-sampling was hand-delivered to the participants to complete before the intervention.

Post-intervention assessment: Soon after the presentation, each participant completed the questionnaire again and received the self-sampling kit.

Self-sample collection: After the post-intervention survey, we gave the sample kits to the women, and provided a secure and private place (bathroom/restroom) at the facility where the presentation took place for the women to collect the sample. Afterward, we mailed the samples to the lab for analysis. Women who were not comfortable taking the sample at the facility were allowed to take the kit home to collect their samples and mailed the kit to the lab. About 90% of the women who completed the self-sample collection conducted the sample collection on the same day of the presentation and at the facility.

### 2.5. Interviews Guide

We conducted one on one face-to-face interviews with 10 women (saturation benchmark was reached) [50] who completed the study to obtain an in-depth understanding of the feasibility, acceptability, and appropriateness scores for the intervention. We also assessed their overall experience with self-sampling and the easiness of using the sample kit. After completing the survey, we invited some of the women for the interviews based on their rating scores for the feasibility, acceptability, and appropriateness survey (i.e., Likert scale 1–5). Our selection criteria were based on women whose ratings were low (1–2), medium (3), and high (4–5). We interviewed women who met the eligibility criteria and collected their samples at the facility. Once the woman was invited, one of the researchers met the woman in a separate conference room and interviewed her while recording the interview. Out of 10 women who participated in the interviews, eight of them completed their sample on the same day of the presentation in the facility. One week after the presentation, we followed up with those women who took the kit home, and if they had already sent in their samples, we met with them at their place of preference, interviewed them, and recorded the conversations. Thus, two women of women who took the sample kit home completed the followed up interview.

We used a 12-item semi-structured interview question to guide the interviews. Some of the interview questions include: “After taking part in the study, (a) why do you think the contents of the presentation were or were not acceptable, feasible, and appropriate to you? (b) Is self-sampling culturally appropriate, acceptable, and feasible for you? Why and why not? (c) Which part of the presentation did you like most and/or you did like most and why? (d) Do you see the 3R model used in the presentation as a better way to promote self-sampling, why and why not? (e) How did you feel about the amount of time spent being part of the study? (f) How did the content of the study influence your decision-making about self-sampling? (g) What are your overall experiences for taking part in the study?”.

### 2.6. Data Analyses

Survey analysis. We utilized descriptive statistics, such as frequencies and means, for the demographic variables and other covariate data. Multivariate logistic regression models were used to analyze the associations between the independent variables and the dependent variable data. We used the paired-sample t-test to analyze the pre-test and post-test data. We use Hayes [51] PROCESS micro in SPSS to conduct the mediation analysis. The significant result was set a priori at a *p*-value < 0.05. All data were analyzed using the SPSS software version 28 (IBM Corp., Armonk, NY, USA).

Interview analysis. All the interviews were audio recorded, transcribed verbatim, and analyzed. We used NVivo to analyze transcribed data. Data were analyzed using thematic coding and content analysis [52]. Two coders (MA and AE) independently read the transcripts and identified common schemes of relevant themes [53]. The Cohen’s kappa < 0.70 (intercoder reliability [54]) agreement was deemed satisfactory.

Mixed methods analysis. We used explanatory sequential mixed methods design data analysis [50]. Survey data were analyzed first, followed by interview data. We integrated both data after separate analyses, developed a table (joint display) that illustrated how the qualitative results enhance the quantitative results, and interpreted the value added by the qualitative explanations.

## 3. Results

### 3.1. Demographic Characteristics

A total of 83 women between the ages of 30 and 65 years (mean age 48.57 ± 11.02 years) completed the intervention. Most (36.14%) of the women were Black/African American, 28.92% of them were Hispanic women, 27.71% were non-Hispanic White, and 7.23% were from other racial/ethnic groups. Almost 70% of the women were not married, and the educational background of the women was fairly distributed across the various degrees. Sixty-eight percent of the women were not working, and 77% of the women reported an annual income of less than $20,000 (Table 1).

Ten of the women (saturation benchmark was reached) who completed the intervention participated in the interviews, and they provided an in-depth understanding of why they found the intervention messages and self-sampling feasible, acceptable, and appropriate.

### 3.2. Self-Sampling Outcomes

Screening behavior and health outcomes. The baseline assessment means score for participants’ readiness to participate in self-sampling was 4.90 ± 2.53 (on a scale of 1 to 10, where 1 indicates less likely and 10 most likely), indicating that the readiness of the women to self-collect sample was very low. After the intervention, 75% (*n* = 62) of the women completed self-testing, 11% (*n* = 7) of them tested positive for high-risk HPV (hr-HPV) genotypes, and 57% of the women with positive results followed up with a care provider. Of the women who tested positive, 57% (*n* = 4) were Black/African American, 14.30% (*n* = 1) Hispanic, 14.30% (*n* = 1) Caucasian, and 14.30% (*n* = 1) others. About 11% of the sample had incomplete results due to the late return of the sampling kit, missing relevant information during the kits registration, and an insufficient sample collected. Twenty-five percent of the women did not participate in self-sampling (Table 1).

Demographic characteristics and screening behaviors. Screening behaviors across the subgroups were analyzed, and the logistics regression results revealed that married women in our study were 3.88 (95% CI, 1.11, 13.59) times more likely to participate in self-sampling behavior compared to unmarried women in the study, after adjusting for the covariates. Additionally, the multivariate logistic regression analysis showed that Black/African American [Adjusted Odds Ratio (AOR) = 0.16, 95% CI, 0.04, 0.65] and Hispanic populations (AOR = 0.12, 95% CI, 0.02, 0.67) were less likely to participate in self-sampling compared non-Hispanic white populations. Women with high school degrees (AOR = 15.97, 95% CI, 2.90, 88.04) and undergraduate degrees (AOR = 4.39, 95% CI, 1.06, 18.19) were more likely to participate in self-sampling compared to women with graduate degrees (Table 2).

Knowledge, attitude, and confidence. We evaluated the intervention effects on the women’s knowledge about, confidence, and attitude toward self-sampling. The paired-sample *t*-test showed that at the baseline assessments, the women’s means score for knowledge about screening was (M = 8.77 ± SD = 3.68), attitude toward screening (M = 27.69 ± SD = 10.89), and perceived confidence about screening (M = 27.77 ± SD = 10.78) were very low. However, at post-intervention assessments, we observed a significant change in women’s means scores for knowledge (M = 11.51 ± SD = 3.50) and attitude (M = 33.81 ± SD = 16.77, *p* < 0.001) after controlling for the baseline scores. There was no significant change in the perceived confidence means score (M = 28.23 ± SD = 11.06, *p* = 0.75).

### 3.3. Mediation Analysis

We determined whether the impact of the intervention on the women’s screening behavior was mediated by TPB constructs (attitude toward behavior and perceived confidence) and knowledge. We conducted mediation analyses using PROCESS. The outcome variable was self-sampling, the predictor variable was the women’s readiness at the baseline, and the mediator variables were post-intervention knowledge, attitude, and confidence. The direct paths (see Figure 1) showed positive relationships between baseline screening readiness vs. post-intervention knowledge (b = 0.5168, se = 0.1463, *p* < 0.05); baseline screening readiness vs. post-intervention attitude (b = 2.4094, se = 0.7031, *p* < 0.001); and post-intervention attitude vs. actual screening behavior (b = 0.0899, se = 0.0355, *p* < 0.01). However, the relationships between screening readiness vs. actual screening behavior (b= −0.0655, se = 0.1709, *p* = 0.7014) and post-intervention knowledge vs. actual screening behavior (b = 2543, se = 0.1392, *p* = 0.0678) were not significant. The indirect effect was tested using non-parametric bootstrapping. The result showed that the relationships between women’s baseline screening readiness and actual screening behaviors were mediated by post-intervention knowledge (IE = 0.1314; 95% CI, 0.0104, 0.4079) and post-intervention attitude (IE = 0.2167; 95% CI, 0.0291, 0.6050), indicating the change in women underscreening before the intervention to the actual screening behavior was indirectly influenced by increased knowledge and positive attitude (Figure 1). Perceived confidence did not influence the women’s screening behavior (it is not shown in the path analysis).

### 3.4. Mixed Method: Survey and Interview Results

The descriptive analysis of the survey results for the feasibility, acceptability, and appropriateness of self-sampling and the 3R model are presented in Table 1 above. The 10 women who completed the interviews found the intervention messages and self-sampling feasible, acceptable, and appropriate. We also assessed knowledge gained through the presentation as well as feedback regarding the self-sampling procedure.

#### 3.4.1. Theme 1: Acceptability

The descriptive analysis showed that over 95% of the women found the intervention content helped them to understand the causes of cervical cancer and influenced their decision to take self-screening. Thus, the intervention met their approval, and it was acceptable. The follow-up interviews explained the women’s high response to the intervention acceptability survey. The women reported that they liked the 3R model because it helped them to be proactive and change their minds about screening. Below are a few quotes from some of the women.

“Reforming and being proactive and how you can change what you have been doing”. “It was educational for me. I like the step-by-step approach to the information presented. The diagram gives a clear picture for me to understand and explain to other people. The presentation is not too long, and it was straight to the point”.(A 33-year-old participant)

#### 3.4.2. Theme 2: Appropriateness

The majority of the women, over 95%, reported that the intervention content (3R messages), activities, and discussion were appropriate and culturally suitable for them. In the follow-up interview, the women indicated that the 3R model intervention was helpful and informative, and it could be used to inform other individuals about cervical cancer and self-sampling. A quote from one of the women explains their overall experience.

“The presentation was clear, precise, and very informative, and I like that [the presenter] asked questions along the way. The information I received today was helpful and as a woman, I have a daughter, I will be able to use the information I learned today to help my daughter when she comes up against it”.(A 39-year-old participant)

#### 3.4.3. Theme 3: Feasibility

From the survey, 100% of the women reported that the intervention content, activities, and time were feasible. In the interviews, the women reported that the use of the 3R format was informative, clear, simple, and well-structured to enhance understanding. The quote below from participants captures the women’s overall experience with the 3R model.

“I will tell others how easy it was and the information I learned, easy to understand and it was relieving to learn those things. I will recommend it to people because I think a lot of people are busy and this sample at-home kit makes it easier for people to do it at home when their lives are fast and chaotic” (A 53-year-old participant). “I will be open to do self-sampling and I believe a lot of women will do self-sampling because they are not comfortable with doctors taking the samples”.(A 44-year-old participant)

However, one woman was not impressed by presenting the information in three different modules. She would have liked to see all the modules combined in one format.

Self-sampling experience: We interviewed the women about their experiences with self-sampling, and their responses showed that they had a favorable experience with self-sampling. The women expressed ease and comfortability collecting the self-sample, and they indicated that comfortability was not present when the physician performed sampling collection, which makes self-sampling more favorable. They reported that the intervention helped them allay fear and anxiety about screening results, make the decision to take a sample, and the instructions provided made it easy to take the sample. A few quotes from the participants summarize the women’s experiences with self-sampling:

“I had no idea about self-sampling but after I learned about it, it is convenient, less embarrassing, unlike going to the actual doctor and lying on the exam table for examination. It is not invasive taking it and it is more comfortable and easier to take it”. (A 34-year-old participant) “The presentation helps me to decide to take the sample because I want to know my status and be educated. I wanted to know if I carry the virus” (A 53-year-old participant). Before taking the sample, I was very nervous that I was going to do this to my body, and I don’t want to do that to my body. After I did it, I found out that it was not difficult at all. It was easy, one, two, three, you are done”.(A 55-year-old participant)

Knowledge: The women expressed that they gained knowledge and awareness about cervical cancer and HPV. Some of the new lessons they learned include the causes of cervical cancer, the fact that HPV infection is a common disease and sexually active individuals are susceptible to contracting the disease, the risk factors, and the preventive tools available. Many additional interview responses showed that some of the women’s misperceptions about cervical cancer and HPV were addressed. Misperceptions such as not knowing HPV could lead to cervical cancer, thinking cervical cancer and HPV was only for younger individuals, thinking that it could not happen to themselves as an individual, and not knowing there were options to help prevent cervical cancer.

“I feel like a learned a lot, just valuable information I didn’t know before about cervical cancer and HPV that I didn’t know and preventative things to be proactive about it” (A 53-year old participant). Wow, I am glad that I took part in the study because I didn’t know anything about the virus and how you can get it. Why nobody has told us anything like this. This is great information to learn” (A 34-year-old participant). “The presentation created awareness for me to know that I may be at risk of having the virus, aware that HPV is so common”.(A 43-year-old participant)

Barriers to self-sampling: We asked the women to tell us about the possible barriers women would face using self-sampling. The challenges identified include the problem of registering the sample kits, the credibility of the self-sampling results, the cost of the kits and lack of health insurance, lack of understanding of the self-sampling procedure, women not feeling comfortable with their own bodies, and fear of knowing that they have the virus. Below are a few quotes from the women:

“Barriers to taking self-sampling could be not understanding what to do and some people are not comfortable with their own body, the cost for self-sampling around $45 can be expensive for some people to buy but compared to doctors’ examination it is less expensive” (A 42-year-old participant). “Some of the barriers can be fear of knowing they have the virus” (A 33-year-old participant). “I don’t see any barriers why any woman wouldn’t want to take it. If women doubt the results, it could be a barrier to take it but to me I will encourage women to take it because it was easier and comfortable to take it. I will recommend it to people to take it”.(A 36-year-old participant)

However, some of the women indicated that self-sampling could alleviate physician-performed cervical cancer screening access (i.e., cost, lack of health insurance, and time) barriers.

“To me, it is easier to use the self-sampling because of the way the economy is, people are being laid off and people are not having insurance or anything. I think self-sampling is good for those who don’t have health insurance because they can’t afford to go to their doctors but can buy the kit and use it at home.” (A 39-year-old participant). “…I think a lot of people are busy and this sample at-home kit makes it easier for people to do it at home when their lives are fast and chaotic”.(A 53-year-old participant)

## 4. Discussion

This study examined the effectiveness of a 3R communication model intervention to increase self-sampling behaviors among low-income women in a small southern U.S. city. The main findings of the study include an increase in screening behavior, positive hr-HPV genotypes, disparities in screening behavior among the subgroups, the acceptability, feasibility, and appropriateness of the intervention, and the change in attitude and knowledge about cervical cancer screening behavior.

The first main finding of this intervention is that the 3R model communication was effective in increasing self-sampling. Both the quantitative and qualitative findings showed that after the intervention, the women had a positive attitude toward screening, and their knowledge about screening increased. We also observed that the increase in the women’s self-sampling behaviors was mediated by the post-intervention increase in knowledge and positive attitude. This indicates there are significant effects of attitude and knowledge that increase screening behaviors of women. As indicated in the Theory of Planned Behavior, an individual’s positive attitude toward a health behavior is associated with a higher likelihood of performing that behavior [55]. Utilizing this 3R communication model, we can further increase the knowledge and attitudes toward self-sampling for cervical cancer. Our findings of increased knowledge and attitude leading to increase sampling behaviors are imperative to this research area. This is supported by other HPV studies indicating that improving knowledge and attitudes toward a health behavior can increase the performance of the health behaviors [56]. Consistent with our study findings, reframing, reprioritizing, and reforming (3Rs) have shown to be effective in addressing stigma and religious objections associated with mammogram usage [44].

Another major finding worth noting is the percentage of women who tested positive for the hr-HPV genotypes. Out of the 75% (*n* = 62) of the women who completed the screening, 11% (*n* = 7) of them tested positive for the hr risk HPV genotypes, suggesting the HPV virus may be prevalent among low-income women. This finding appears to agree with other studies that show high detection rates of cervical cancer among women with low socioeconomic status [10], and among uninsured and Medicaid-insured persons [10,12,13].

Additionally, our findings show that the women found the 3R communication intervention acceptable, feasible, and appropriate. The women seemed to appreciate the simple and structured nature of the 3R communication model. In their own words, the women described the model as “simple and easy to understand” and “it is also informative”. Furthermore, the 3R model seems to be effective in addressing misconceptions and demystifying myths about cervical cancer and cervical cancer screening. Further studies need to be conducted to validate these findings in a large sample size so as to enhance the adoption and implementation of the model at the population level.

Another strong finding was that the women reported a positive experience with collecting self-sampling. This finding is supported by a recent meta-analysis of 37 studies among 18,516 women from 24 countries across five continents and a scoping review [57] that indicated women have a strong acceptance and preference for self-sampling over clinician sampling [37]. The women in our study implied that self-sampling could help alleviate the economic burden and increase the accessibility of cervical cancer screening for low-income women. This finding lends credence to other studies that concluded that self-sampling might be a suitable alternative method for low-resource settings or among patients reluctant to undergo pelvic examinations [39,40,41].

The women in our study identified several challenges that could be barriers to self-sampling. They include problems registering the sample kits, doubt about the self-sampling results, socioeconomic factors such as cost and lack of insurance, problems performing self-sampling, feeling uncomfortable collecting samples, and fear of positive results. A study by Pierz et al. [58] identified comparable self-sampling barriers, which include a lack of education about the self-sampling procedure, feeling uncomfortable, embarrassed, or in pain from the self-collection procedure, fear of consequences and perceived competence about the ability to self-collect.

Limitations: The study aimed to evaluate the effectiveness of the intervention and understand further contextual factors through qualitative responses. Though the insights we gained from this study were valuable, the findings should be interpreted with caution because of potential limitations. For instance, there may be a lack of generalizability due to a relatively small sample. However, this study has the potential to be scaled up to a broader and larger sample in the future. Additionally, the face-to-face method of collecting the qualitative data may have presented social desirability bias in the responses of the women. To combat this, the responses were collected with the intention that they would remain anonymous and not identified by the participant. Lastly, the single-group pre-test-post-test quasi-experimental design of this study is a potential limitation. Due to the nature of this design, there is the ability that external, non-intervention effects account for the improvement of the post-test scores. The researchers do believe these effects were limited since the time lapse between the pre-test, and post-test was very minimal, thus restricting the opportunity for external effects to confound the post-test scores. With additional resources in the future, it would be beneficial to conduct this study with a multi-group design.

Despite these limitations, there are many strengths to this study. First, to the best of the author’s knowledge, this is one of the first to study the effectiveness of the 3R communication model in increasing cervical cancer self-sampling behaviors in this population. Second, the mixed-methods approach to this study provides valuable insight into why this intervention was effective based on input from the participating women. Finally, this intervention can be scaled up and administered to larger populations to aid in increasing cervical cancer screening.

## 5. Conclusions

The findings of this study indicate the effectiveness of a 3R communication model intervention in increasing self-sampling behaviors among low-income women. The implications of this study include scaling up and increasing the reach of this intervention to include larger populations of women from additional regions. This intervention is essential to help increase cervical cancer screening in women to detect cancer at earlier stages and ultimately aid in decreasing cervical cancer deaths of women in low-income communities.

## Figures and Tables

**Figure 1 healthcare-11-01323-f001:**
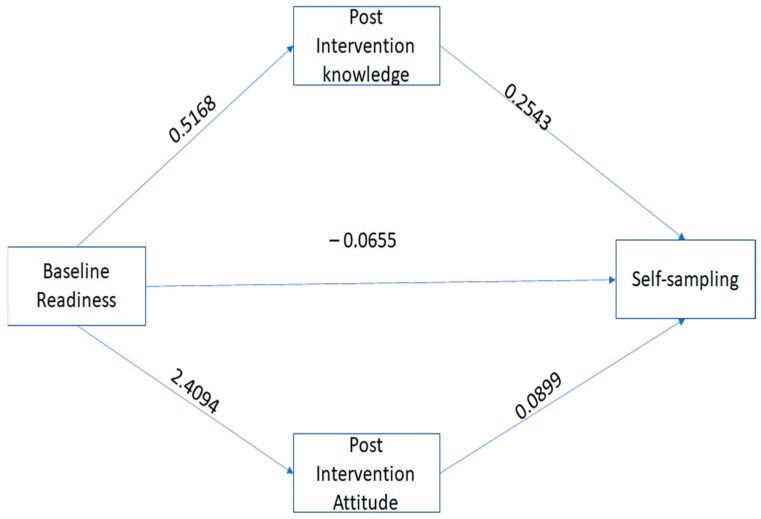
Path analyses showing that knowledge and attitude mediate women’s screening behavior.

**Table 1 healthcare-11-01323-t001:** Demographic characteristics and other covariates of the participants (*n* = 83).

	Frequency	Percent (%)
Age		
Age range 30–65; Mean (SD) = 48.57 ± 11.02		
Race/Ethnicity		
Caucasian	23	27.71
Black or African American	30	36.14
Hispanics	24	28.92
Other	6	7.23
Marital Status		
Not married	58	69.88
Married	25	30.12
Education		
Graduate degree or higher	17	20.48
Undergraduate	14	16.87
High School	25	30.12
Less than High School	27	32.53
Insurance		
No	30	36.14
Yes	53	63.86
Employment		
Not working	48	57.83
Working	35	42.17
Annual Income		
<$20,000	64	77.11
>$20,000	19	22.89
Screening behavior		
Did not screen	21	25.30
Screened	62	74.70
Screening outcomes		
Incomplete	8	12.90
Negative	47	75.58
Positive	7	11.29
Acceptability		
Not acceptable	4	4.82
Acceptable	79	95.18
Appropriateness		
No appropriate	4	4.82
Appropriate	79	95.18
Feasibility		
Not feasible	0	0.00
Feasible	83	100.00

**Table 2 healthcare-11-01323-t002:** Bivariate association and multivariate models of correlates of screening behavior by selected demographic characteristics (*n* = 83).

	Unadjusted OR (95%, CI)	Adjusted OR (95%, CI)
Age		
30–40	1.47 (0.38–5.66)	1.48 (0.42–5.24)
41–50	2.2 (0.54–9.01)	1.36 (0.36–5.14)
>50	Ref (--)	Ref (--)
Marital Status		
Married	3.82 (1.13–12.94	3.88 (1.11–13.59)
Not Married	Ref (--)	Ref (--)
Insurance		
Yes	1.08 (0.40–2.96)	1.12 (0.39–3.23)
No	Ref (--)	Ref (--)
Employment		
Working	1.18 (0.45–3.11)	1.08 (0.40–2.94)
Not working	Ref (--)	Ref (--)
Income		
Yes	0.86 0.26–2.82)	0.96 (0.27–3.43)
No	Ref (--)	Ref (--)
Race/Ethnicity		
Other	0.93 (0.10–8.46)	0.84 (0.12–5.96)
African American	2.78 (0.78–9.85)	0.16 (0.04–0.65)
Hispanic	4.27 (1.01–18.11)	0.12 (0.02–0.67)
Non-Hispanic white	Ref (--)	Ref (--)
Education		
Less than high sch	1.23 (0.24–6.45)	1.35 (0.24–7.46)
High school	0.32 (0.07–1.50)	15.97 (2.90–88.04)
Undergraduate	0.15 (0.03–0.85)	4.39 (1.06–18.19)
Graduate	Ref (--)	

## Data Availability

The data presented in this study are available on request from the corresponding author. The data are not publicly available due to privacy.

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
