# Peer review of "Intervention to Increase Cervical Cancer Screening Behavior among Medically Underserved Women: Effectiveness of 3R Communication Model"

_healthcare, 2023, doi:10.3390/healthcare11091323_

Round 1

Reviewer 1 Report

The study was well done and the  article is well written.  The only suggestions are:   (1) to explain how the snowball sampling method was done - what questions were asked?  Was a face-to-face recruit asked to name others who were between 30 and 65 years of age and had never had a pap smear  or HPV test.....;(2) how specifically was TPB operationalized in the development of the Modules?; (3) what pedagogy was used in delivering the Modules?   lecture?  power points?  handout?  etc; and (4) how were the 10 women identified for the face-to-face interviews?

Reviewer 2 Report

Thank you for submitting the manuscript to the journal "Healthcare." Manuscript is quite well written and only several revisions are needed to accept it.

Firstly, using 3R framework is good to connect your training program to the educational framework. However, since a participants gave you a feedback that the educational intervention was a bit too long. It took 30-45 minutes to include Modules 1-3. It might be too long for this content. Please discuss the time cost effectiveness of this program compared to the volume of the contents.

Lastly items of the Survey are not listed in the manuscript. Please provide the following information: how many items were included, reliability of the results for whole items and for each category. 

Reviewer 3 Report

Dear authors, it is an interesting article, the use of mixed methods increase the value of the article and allow to have an overview about barriers and outcomes of an educative program self-sampling implementation

Some general suggestions

Line 50 Suggest to include Mortality crude rate per 100000

Line 76 Use the same definition in other part of the document is Co-testing  

Intervention Delivery

What do you use for face-to-face presentation? Roll Up, Power point or just was a lectura

Survey.

The questionary was validated somewhere perhaps include is feasible to include the questionnaire as additional material or describe briefly the content and the questions  

Interviews guide

Is not clear how this qualitative part was done. Is also important to briefly describe the content of the guide. Who guide the interview, have you preformed a content analysis?  Do you use the same selection criteria for participants? It was a FDG or in on deep interview? Where do interview took place? There was only one person leading the interview?  

Data Analyses

How many time after was the post-test made, how do you contact participants if was some time after

Demographic Characteristics

Line 273 All women participate in the qualitative analysis?

Results

Is important to know age of participants and to which FDG belongs or which participant is. I some quotations are described  

Is not clear if the same participants are involved on qualitative and quantitative analysis  

Round 2

Reviewer 3 Report

The authors have addressed all my suggestions 

I have no additional comments